# Insight into the Exemplary Physical Properties of Zn-Based Fluoroperovskite Compounds XZnF_3_ (X = Al, Cs, Ga, In) Employing Accurate GGA Approach: A First-Principles Study

**DOI:** 10.3390/ma15072669

**Published:** 2022-04-05

**Authors:** Anwar Habib, Mudasser Husain, Muhammad Sajjad, Nasir Rahman, Rajwali Khan, Mohammad Sohail, Ismat Hassan Ali, Shahid Iqbal, Mohammed Ilyas Khan, Sara A. M. Ebraheem, Ahmed M. El-Sabrout, Hosam O. Elansary

**Affiliations:** 1Department of Physics, Kohat University of Science & Technology, Kohat 26000, Pakistan; anwarhabib854@gmail.com; 2Department of Physics, University of Lakki Marwat, Lakki Marwat 28420, Pakistan; mudasserhusain01@gmail.com (M.H.); rajwali@ulm.edu.pk (R.K.); msohail@ulm.edu.pk (M.S.); 3Department of Chemistry, College of Science, King Khalid University, Abha 62529, Saudi Arabia; ihali@kku.edu.sa; 4Department of Physics, Albion College, Albion, MI 49224, USA; siqbal@albion.edu; 5Department of Chemical Engineering, College of Engineering, King Khalid University, Abha 62529, Saudi Arabia; mkaan@kku.edu.sa; 6Department of Chemistry, College of Science and Arts, King Khalid University, Saratabida 61914, Saudi Arabia; samibrahim@kku.edu.sa; 7Department of Applied Entomology and Zoology, Faculty of Agriculture (EL-Shatby), Alexandria University, Alexandria 21545, Egypt; elsabroutahmed@alexu.edu.eg; 8Plant Production Department, College of Food and Agriculture Sciences, King Saud University, Riyadh 11451, Saudi Arabia; helansary@ksu.edu.sa; 9Floriculture, Ornamental Horticulture, and Garden Design Department, Faculty of Agriculture (El-Shatby), Alexandria University, Alexandria 21545, Egypt; 10Department of Geography, Environmental Management, and Energy Studies, University of Johannesburg, APK Campus, Johannesburg 2006, South Africa

**Keywords:** structural, electronic, optical and elastic properties, Wien2K

## Abstract

Using the full-potential linearized augmented plane wave (FP-LAPW) method, dependent on density functional theory, the simple cubic ternary fluoroperovskites XZnF_3_ (X = Al, Cs, Ga, In) compound properties, including structural, elastic, electronic, and optical, are calculated. To include the effect of exchange and correlation potentials, the generalized gradient approximation is applied for the optimization operation. This is identified, when we are changing the metallic cation specified as “X” when shifting to Al from Cs, the value of the bulk modulus is found to increase, showing the rigidity of a material. Depending upon the value of the bulk modulus, we can say that the compound AlZnF_3_ is harder and cannot be compressed as easily as compared to the other three compounds, which are having a lower value of the bulk modulus from AlZnF_3_. It is also found that the understudy compounds are mechanically well balanced and anisotropic. The determined value of the Poisson ratio, Cauchy pressure, and Pugh ratio shows our compounds have a ductile nature. From the computation of the band structure, it is found that the compound CsZnF_3_ is having an indirect band of 3.434 eV from (M-Γ), while the compounds AlZnF_3_, GaZnF_3_, and InZnF_3_ are found to have indirect band gaps of 2.425 eV, 3.665 eV, and 2.875 eV from (M-X), respectively. The optical properties are investigated for radiation up to 40 eV. The main optical spectra peaks are described as per the measured electronic structure. The above findings provide comprehensive insight into understanding the physical properties of Zn-based fluoroperovskites.

## 1. Introduction

Among various compounds, one of the families of compounds, known as perovskites, has drawn the attention of material scientists in the field of science and technology. These kinds of materials are studied by researchers using computational and experimental approaches. The compound (CaTiO_3_), which is the mineral perovskite, was firstly investigated in [1]. After this, incredible attention was given to this family of compounds. Due to having effective uses in science and technology, an extraordinary concentration was set to optical, as well as the electronic properties of perovskites, in the industry of lenses and semiconductors [2]. ABX_3_ is the representative stoichiometry of perovskite crystal structure, where atoms “A”and “B” are from a standpoint for the cations, which are well-specified for an alkali metal andan alkaline earth metal, respectively, while “X” stands for an anion. Due to atom X, several forms occur from the general structures of perovskite, the compound that would be fluoroperovskites (ABF_3_), if fluorine appears in the place of X molecule. Whenever oxygen is available, it then becomes oxides-perovskite (ABO_3_) and nitride-perovskite (ABN_3_), when nitrogen is positioned at the X atom site. The compounds of fluoroperovskites are most often divided into two sections, the first of which is recognized, such as oxide perovskites, when oxygen is selected in the place of anion, and, as we take anion from the halogens group, that is why the second one will be called halide perovskites. In perovskite materials, all types of compounds exist, such as conductors, semiconductors, and insulators. Due to its well-balanced structure, the exceptional properties of perovskite compounds are commonly analyzed using different experimental methods and modeling. The compound of fluoroperovskites is fascinating for its dielectric examinations, and the application in solid-state devices for this type of material is mostly electronic, optical, etc. The optical properties of perovskite compounds, though, are anisotropic and show the influence of birefringence, plus their structure is compatible with chemical composition, temperature, and pressure [3]. Lufaso and Woodward have previously researched how cubic perovskites could be converted to another structure of the crystal [4]. The ternary fluorides of perovskite crystal structure have been broadly studied for several years; due to their optical characteristics [5,6], these materials have several practical uses such as thermo-electricity [7], as well as physical properties, such as ferroelectricity [8], anti-ferromagnetism [9], and semi-conductivity [10]. In the medical field, such types of materials are significant, being utilization during radiation therapy as well as in X-rays and gamma rays, and in the imaging plates of thermal neutrons; thus, such compounds have a distinct medical application. Additionally, the usage of fluoroperovskite materials in numerous scientific applications, as a fuel for cells, memory appliances, photovoltaics, etc., has been studied and has been demonstrated to be outstanding compounds for microelectronics, as well as in telecommunications [11,12,13]. The crises of energy are reduced by the outstanding thermoelectric properties of these materials and can help the people, where there is a high need for energy. To find out about the material of the form of fluoroperovskites, researchers have made efforts to explore these types of compounds. Based on the Seebeck coefficient and thermal conductivity, the thermoelectric properties of such materials have attracted the attention of the community of technology. Sabir et al. investigated a few of these compounds [14]. The fluoride compounds have a wide bandgap and are known to be a better candidate. The creation of complex material, with a crystal structure that fits the lattice and large band gaps of fluorides, offers the chance order to be alloyed, which follows the lattice as well as the engineering of the band gaps [15]. Initial studies, which are experimental or theoretical, are based on how fluorine and organic or inorganic elements can be mixed with transition metal (TM) [16,17,18,19,20]. Such combinations produce excellently balanced compounds of fluoroperovskites, which have been studied to make these compounds thermodynamically and mechanically stable [21,22]. Due to the diversity of structural variations and configurations, transition-metal-type fluoroperovskites have interesting magnetic properties. Perovskites with a cubic structure, such as the compound KMF_3_ (M = Mn, Ni), exhibit an anti-ferromagnetic order at higher temperatures [23]. Such compounds also have several suitable thermoelectric and optical applications.

In this investigation, our purpose is to provide the structural, elastic, electronic, and optical properties of fluoroperovskite XZnF_3_ (X = Cs, In, Ga, Al) compounds, by applying the GGA approach. This technique of treating the electronic correlation, with the model such as that of Hubbard, was suggested in the 1990s in density functional theory.

## 2. Methodology

An ab initio full-potential linear augmented plane wave (FP-LAPW) method, within the generalized gradient approximation (GGA) and as implemented in the WIEN2K code [24], is used. The expansion of spherical harmonic is applied. The plane–wave basis set is selected in the interstitial region (IR) of the unit cell inside the spheres that do not overlap with the muffin-tin radius (RMT). For such compounds, the value of RMT is chosen in such a way that the spheres have not overlapped. To obtain the total energy convergence, the basic functions in the IR are expanded up to R_mt_ × K_max_ = 6.0 inside the atomic spheres for the wave function. For structural optimization, integral through the Brillouin zone is completed, taking 2000 k-points from the mesh in the full Brillouin zone. The structural properties are simulated using the Birch–Murnaghan equation of state, by fitting the energy versus volume curve of the crystal unit cell. The electronic properties are investigated using the GGA approximation within the high symmetry points of the first Brillouin zone. In the real system, the electron density is not as uniformly distributed as the ideal free electron gas. Therefore, a gradient correction of charge density is introduced, called generalized gradient approximation (GGA). Under the condition of GGA, the exchange-correlation energy *E_xc_*[*ρ*] is the function of not only the electron density *ρ*(*r*) but also the electron-density gradient ∇ρ. Under the GGA, the exchange-correlation effect also decomposes into two parts, the exchange energy and correlation energy, which are calculated in their appropriate functional forms. GGA can correct the overestimated binding energy of LDA in molecules and solids as well as extend the processing system to the energy and structure of the hydrogen-bond system. The approximation greatly improves the calculation results of the energy related to electrons and exchange. DFT, based on functional GGA, has achieved great success in the efficiency and accuracy of electronic structure calculations and has become the most popular calculation scheme in multi-electronic systems [25,26,27]. The DOS is reported within the energy ranges from −8 eV to 8 eV. For the computation of elastic properties, the IRelast package developed by Murtaza is employed for elastic constants and other parameters. The optical properties for all the interesting compounds are investigated, using the dielectric function within the energy range from 0 eV up to 40 eV. Based on the aforementioned computational methods, we found very accurate and precise results, as described below.

## 3. Results and Discussion

### 3.1. Structural Properties

The ideal cubic fluoroperovskite structural compounds, ABF_3_ has a space group of Pm-3m (221). Among perovskite compounds, we choose to study XZnF_3_, which crystallizes in the simple cubic structure, having Wyckoff positions of (0, 0, 0), (0.5, 0.5, 0.5), and (0, 0.5, 0.5) for X, Zn, and F, respectively, as shown in Figure 1.

We examine the structural properties of the fluoroperovskite compounds XZnF_3_ (X = Al, Cs, Ga, In), using a common simulation program, Wien2k. By analyzing energy E_0_, ground state volume V_0_, lattice constants that are optimized, bulk modulus B, and the bulk modulus pressure derivative $B^{\prime }$, the structural properties of the under-studied materials are taken into consideration. Using the volume optimization process, all these parameters are determined, which adjusts the whole system. To include the effect of exchange and the correlation potential, the generalized gradient approximation along with the term is applied for the optimization of the operation. To identify different characteristics of specific materials, the role of this operation is very significant. Due to this approach, the volume versus energy parabolic curve, called the curve of optimization, is achieved, using the Birch–Murnaghan state equation [28], by adjusting several states’ energy values against their related volume. In Figure 2, the curve between volume and energy is shown for the compounds XZnF_3_ (X = Al, Cs, Ga, In). These graphs show that, initially, as the unit cell’s energy slowly rises, its volume reduces, and if the energy of the unit cell reaches its lowest limitation, the value of the energy at this lowest limitation is referred to as energy at the ground state, whereas the volume value of the unit cell at the energy of this ground state is called the volume at the ground state.

Table 1 provides the specified value for all the structural parameters. It can be seen from the table that when shifting to Al from Cs, the value of the bulk modulus is found to increase by changing the metallic cation described by ‘X’, showing a material’s rigidity. It can, therefore, be inferred that AlZnF_3_ is harder and less compressible, based on the B_0_ values more than CsZnF_3_, as displayed in Table 1. Although we did not observe sufficient conclusive experimental or theoretical findings, we confirmed our results were in line with the AFLOW’s accessible data [29]. Similar results for the structural properties are reported by Husain et al., while investigating the different physical properties of the *NaQF_3_* (Q = Ag, Pb, Rh, and Ru) compounds. The structural stability, bulk modulus, lattice constants, and other ground state parameters were reported by them. Based on the comparison of the interesting compounds with another similar type of fluoroperovskites, we can confidently declare that our selected materials have a stable cubic crystal structure.

### 3.2. Elastic Properties

Few of the elastic constants (C_ij_) are found that exhibit the properties of a material to apply forces. Such elastic constants (C_ij_) are significant and essential to describe the mechanical response of compounds. It relates to the mechanical and dynamic features of crystals, whereas elastic constants further characterize the deformation of material as stress is employed, and when stress is eliminated, it comes back to its normal form [30]. Details about the stability of the structure, the bonding behavior of the neighboring planes of atoms, and the character of anisotropy could also be acquired using elastic constants. The nature of each of these compounds in this work is cubic, so C_11_, C_12_, and C_44_ are the three different elastic constants that were introduced in the cubic crystal symmetry to quantify the mechanical characteristics like rigidity, and also stability. Therefore, those three constants are enough to explain in detail the elastic properties, such as the mechanical response and stability of the structure of the material. Materials having a crystal structure in the cubic form, would be well balanced mechanically when the formulas of restriction shown below are satisfied.
(1)$$\begin{array}{c} C_{44}>0,\\ C_{11}-\frac{C_{12}}{2}>0\\ C_{12\;}<B\;<C_{11};\\ {\;C}_{11}<B\;<C_{11};\\ \left(2C_{11}+{\;C}_{11}\right)/3>0\; \end{array}$$

The conclusions drawn from the compounds investigated indicate that all XZnF_3_ (X = Al, Cs, Ga, In) compounds obey all of the stability criteria described above, thus, we can say that our compounds are stable mechanically. Table 2 displays the values defined for C_11_, C_12_, and C_44_ for AlZnF_3_, CsZnF_3_, GaZnF_3_, and InZnF_3_, applying GGA together with others estimated values. To the best of our knowledge, C_ij_ values for these compounds have not been reported to match our findings.

It could be seen from the table that the C_11_ value for GaZnF_3_ is greater, which indicates that it is more elastic than other materials. The higher C_44_ value gives information on the stiffness of a material. The compound CsZnF_3_ could be stiffer, from the measured C_44_ values, than the other three compounds, and the elastic anisotropy factor (A) of a compound is an essential factor in the applied and engineering sciences. It provides an idea that a micro-crack is expected in the induction of compounds. This could also be applied to describing the degree of elastic anisotropy. It is said that the material is like a full isotropic, whenever the elastic anisotropy factor [A= 2C_44_/(C_11_ − C_12_)] is 1. This material is also called elastic anisotropic. The elastic anisotropy value is determined by applying the GGA approach to the XZnF_3_ compounds (X = Al, Cs, Ga, In), shown in Table 2. From the table, it is clear that the value of A is not equal to 1 for all compounds, so our compounds are anisotropic. The Paugh ratio suggested by B/G is just another parameter used to measure ductility, and this ductility has a threshold value of 1.75. Table 2 shows defined values for XZnF_3_ (X = Al, Cs, Ga, In). It explains that for all of the compounds tested, the B/G values are higher than 1.75 and, therefore, show ductile nature. Poisson’s ratio (v) also defines the ductility of the compounds and their brittleness; the poison ratio threshold value is 0.26, so a compound having a Poisson ratio greater than 0.26 is ductile, whereas a compound having a threshold value lower than 0.26 is brittle. The value of the poison ratio (v) is higher than 0.26 for all compounds under study, so the essence of all these materials is ductile. Material elasticity is also indicated by Cauchy pressure, indicated by C_P_. If the C_P_ value is positive, it will be ductile, and if negative, then the compound will be brittle. It is given by this equation:(2)$$C_{P}={\;C}_{12}-{\;C}_{44}$$

From Table 2, it is clear that the C_P_ value for all materials is positive, so this implies a ductile nature. Table 2 also shows certain elastic parameters that represent the rigidity of the material, including Shear Modulus (G), Bulk Modulus (B), and Young Modulus (E).

### 3.3. Electronic Properties

The structures of the electronic band provide the main significance in solid-state physics while investigating different materials. This defines the role of the states of energy available at different symmetric points. Various properties of the materials are known to be dependent on band structure, such as magnetic, electrical, and thermal characteristics. Information about a material, whether it is used for conductors, semiconductors, or insulators, is obtained from the band structure. The level of the fermi energy, described by the dotted lines, is highly meaningful in the band structure depicted by E_F_, which is filled at zero temperature. The part higher than the Fermi level is called a conductive band, while the portion just below the Fermi level is called the valence band, and the lowest part of the valence band is called the core band. The position of the Fermi level makes a major contribution to the essence of a compound. If this is situated between the conductive band and the valence band, the material will be a semiconductor or an insulator, otherwise the compound would be taken as metallic. Dependent on such concepts, we calculated the structure of the electron band by applying GGA, assuming 2000 k-points from the mesh to implement the exchange as well as the correlation effects. The calculated electronic band structure for the compounds XZnF_3_ (X = Al, Cs, Ga, In) is shown in Figure 3. It can be seen that the lower point for CsZnF_3_ of the conduction band is situated at the Γ point of symmetry, whereas the higher point is at the M point of symmetry in the valence band, resulting in an indirect bandgap of 3.434 eV (M-Γ). The InZnF_3_, GaZnF_3_, and AlZnF_3_ compounds are indirectly band gaps of (M-X), having band gaps of 2.875 eV, 3.665 eV, and 2.425 eV, respectively.

We will now take a glance at the available states for electrons inside the system. State density gives a very complete description of the compounds and explains the valence band and the conduction band’s involvement, larger and smaller. This includes both the valence and conduction band structure as well as explanations of state distribution, which suggests what consists of the valence band state and the conduction band. Knowledge about the electron density of state (DOS) is needed to understand and explain the bonding nature, band structures, etc., of a compound. To estimate certain characteristics of XZnF_3_ (X = Al, Cs, Ga, In), materials are shown in Figure 4, and the total (TDOS) and partial density of state (PDOS) are calculated. The state density plot for AlZnF_3_ reveals that there is a great contribution of the F-p state in the valence band, along with a little bit of contribution from the state of Al-s and Zn-d, whereas there is a minor contribution from the F-p state in the conduction band. Looking at the CsZnF_3_ plot, it is evident that there is a significant contribution from the Zn-d and F-p states in the conductive band, along with a contribution from the Cs-p state. In the case of GaZnF_3_ and InZnF_3_, there is a dominant contribution from the F-p and Zn-d states in the valence band, while in the conduction band there is a very minor contribution from the F-p state, a slight contribution from the Ga-s state (in the case of GaZnF_3_), and an In-s state (in the case of InZnF_3_) could be seen in the valence band.

### 3.4. Optical Properties

It is particularly important for the study of the optical properties of a compound because it describes in what way electromagnetic waves react to a compound. It also describes the internal components of the compound. The optical properties of XZnF_3_ compounds (X = Al, Cs, Ga, In) are evaluated by using the GGA method within the Density Functional theory scheme. A clear explanation of all-optical parameters, consisting of the real part, the imaginary part of the dielectric function, reflectivity, refractive index, extinction coefficient, absorption coefficient, and optical conductivity, is provided below. From the information of the complex dielectric function $\varepsilon \left(\omega \right)=\varepsilon_{1}\left(\omega \right)+i\varepsilon_{2}\left(\omega \right)$, the optical characteristics of matter could be defined [31,32,33]. The real and imaginary parts of the dielectric function, respectively, are $\varepsilon_{1}\left(\omega \right)$ and $\varepsilon_{2}\left(\omega \right).$ The imaginary part $\varepsilon_{2}\left(\omega \right)$ is directly connected to the structure of the electronic band. Outlining all relevant transitions amongst the occupied and the unoccupied states, takes into account sufficient transitional dipole matrix elements with the expression below [34,35,36].
(3)$$\varepsilon_{2}\left(\omega \right)=\frac{e^{2}\hbar }{\pi m^{2}\omega^{2}}\sum_{v,c}\int {\left|M_{cv}\left(k\right)\right|}^{2}\delta \left[\omega_{cv}\left(k\right)-\omega \right]d^{3}k$$

The real part could be derived from the imaginary part, utilizing the famous Kramers–Kronig transformation [37].
(4)$$\varepsilon_{1}\left(\omega \right)=1+\frac{2}{\pi }\;P\;\int_{0}^{\infty }\frac{\omega^{\prime }\varepsilon_{2}\left(\omega^{\prime }\right)}{\omega^{’2}-\omega^{2}}d\omega^{\prime }$$
where *P* sets out the fundamental value of the integral. The other optical parameters such as the optical conductivity, reflectivity, coefficient of absorption, extinction coefficient, and refractive index can be obtained by using the real and imaginary part of the dielectric function. It can be seen from Figure 5 that the compounds are more effective in the range of energy of 2.5 eV to 9 eV, in the case of $\varepsilon_{1}\left(\omega \right),$ while in the case of $\varepsilon_{2}\left(\omega \right)$ the compounds are more active from 4 eV to 14 eV, since in between this energy range there are many sharp peaks, which arise due to the electronic transition from the valence band to the conduction band. The threshold energy of the dielectric function is occurring at about 2.425 eV, 3.434 eV, 3.665 eV, and 2.875 eV for AlZnF_3_, CsZnF_3_, GaZnF_3_ and InZnF_3_, respectively. These energy points are called the basic absorption edge. At the range of high energy, it could be seen that the optical activity is lower than 1 eV.

#### 3.4.1. Refractive Index and Extinction Coefficient

An essential parameter that specifies how much the refractive index refracts. This is very useful in photoelectrical applications. This has two parts, real and imaginary part. The imaginary part specified by K(ω) is known as the coefficient of extinction. The index of refraction value is different for various materials. The defined refraction index for compounds XZnF_3_ (X = Al, Cs, Ga, In) can be seen in Figure 6. For AlZnF_3_, CsZnF_3_, GaZnF_3_, and InZnF_3_ compounds, the static refractive index represented by n(0) is 2.0357, 1.652, 1.878, and 1.914, respectively. The highest value of the refraction index for AlZnF_3_, CsZnF_3_, GaZnF_3_, and InZnF3 is 2.707 at 2.855 eV, 2.009 at 12.17 eV, 2.492 at 4.693 eV, and 2.487 at 3.161 eV, respectively, as depicted in Figure 6. Looking at the extinction coefficient K(ω) shown in Figure 7, it can be seen from the figure that for AlZnF_3_, CsZnF_3_, GaZnF_3_, and InZnF_3_, the highest medium absorption is at 5.857 eV, 15.84 eV, 7.022 eV, and 6.409 eV, respectively.

#### 3.4.2. Absorption Coefficient

The absorption coefficient displayed as I(ω) defines the absorption of light through an optical medium per unit length. This determines how far it pierces inside the medium. A compound having a greater coefficient of absorption has a strong capacity of absorption and a compound with a poor absorption coefficient is poorly absorbent and clear for incident light. The absorption coefficient for compounds XZnF_3_ (X = Al, Cs, Ga, In) is indicated in Figure 8. The threshold point of the I(ω) is practical energy, so compounds start absorbing electromagnetic radiation quickly. For AlZnF_3_, CsZnF_3_, GaZnF_3_, and InZnF_3_, it is located around 2.549 eV, 4.203 eV, 3.713 eV, and 3.039 eV, respectively. The absorption intensity of the spectrum I(ω) keeps fluctuating. At about 23.32 eV for AlZnF_3_, 19.460 eV for CsZnF_3_, 19.2 eV for GaZnF_3_, and around 18.97 eV for InZnF_3_, the maximum I(ω) is calculated to be about 104.832, 234.034, 133.71, and 149.47, respectively. After the maximum point, a sharp reduction in the spectrum occurs at a high energy level.

#### 3.4.3. Reflectivity

The coefficient of reflectivity or reflection refers to the ratio of the power that is reflected in the incident power, which is expressed through R(ω). Figure 9 depicts the plots of reflectivity for XZnF_3_ (X = Al, Cs, Ga, In). The optical properties perform a significant part in the analysis of the structure of compound surfaces. High reflectivity is observed for AlZnF_3_, CsZnF_3_, GaZnF_3_, and InZnF_3_, as show in Figure 8, at about 27.188%, 45.22%, 36.55%, and 35.15%, respectively. In the energy range of [(9.779−13.027 eV), (16.948–23.25 eV), (23.99−35 eV)] for AlZnF_3_, [(10.085–13.701), (23.137−35)] for CsZnF_3_, [(0−5.735 eV), (25.281−35 eV)] for GaZnF_3_, and [(9.901–18.480 eV), (23.137−35 eV)] for InZnF_3_, the R(ω) is lower than 10%. In these areas, for incident photons the testing compounds are clear, suggesting that a compound may be utilized to manufacture lenses and anti-reflective coating. Current research reveals that reflectivity hits a maximum value of 27.188% at about 3.835 eV for AlZnF_3_, 45.22% at about 21.482 eV for CsZnF_3_, 36.55% at about 6.960 eV for GaZnF_3_, and 35.15% at around 6.53 eV for InZnF_3_, respectively.

#### 3.4.4. Optical Conductivity

The optical conductivity $\sigma \left(\omega \right)$ is a significant optical design parameter, which describes the electron’s conduction through an electromagnetic field that is employed. Figure 10 shows the investigated $\sigma \left(\omega \right)$ for XZnF_3_ compounds. The key point of the $\sigma \left(\omega \right)$ spectrum is the functional energy that significantly increases the conduction of electrons. For AlZnF_3_, CsZnF_3_, GaZnF_3_, and InZnF_3_, this is determined around 2.426 eV, 4.0196 eV, 3.5906 eV, and 2.977 eV, respectively. The higher $\sigma \left(\omega \right)\;$is found to be 3869.75 Ω^−1^ cm^−1^ at 5.674 eV for AlZnF_3_, 7725.49 Ω^−1^ cm^−1^ at 13.578 eV for CsZnF_3_, 6577.03 Ω^−1^ cm^−1^ at 6.899 eV is for GaZnF_3_, and 5834.73 Ω^−1^ cm^−1^ at 6.654 eV for AlZnF_3_.

## 4. Conclusions

We have concluded the structural, elastic, electronic, and optical properties implying DFT for the cubic fluoroperovskite XZnF_3_ (X = Cs, In, Ga, Al) compounds. From our results, these compounds are structurally optimized, which is achieved using the Birch–Murnaghan state equation. From these optimization curves, we have found these compounds are more stable when having optimized energy that concerns the corresponding volume. Mechanically, all our compounds are found to be stable, ductile, and anisotropic. The electronic properties of these compounds reveal that the compounds of interest are semiconductors, having an indirect bandgap in CsZnF_3_ from (*Mv* − *Γ*), while InZnF_3_, GaZnF_3_, and AlZnF_3_ have an indirect bandgap (*Mv* − X). We have found a bonding character for these compounds that is dominantly ionic and partly covalent. Through the investigation of optical characteristics of the dielectric function from the real and the imaginary parts, we have found that these compounds possess high optical conductivity and absorption at low energy. We are fully confident in our more precise results, and the applications of the above-reported compounds can be deemed relevant in many electronics and semiconducting processing industries.

## Figures and Tables

**Figure 1 materials-15-02669-f001:**
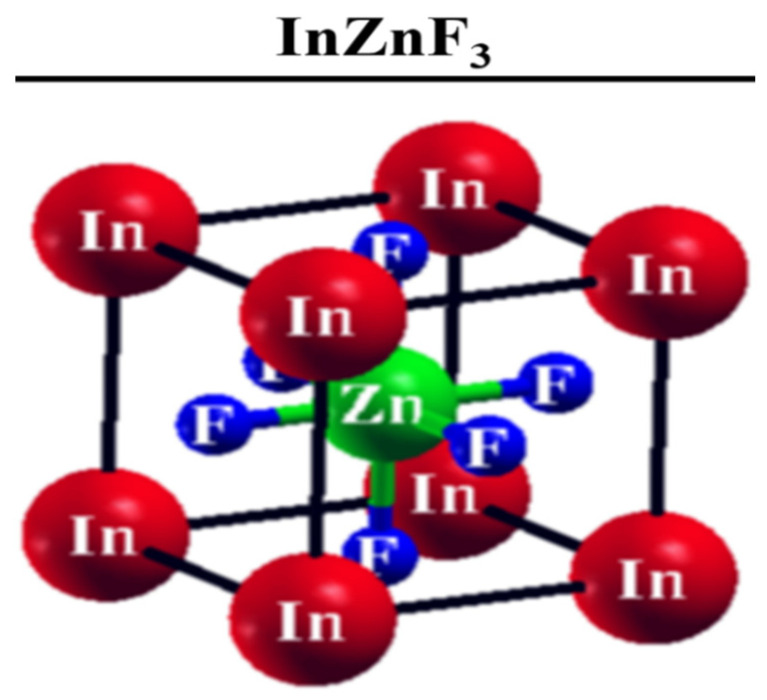
Prototype crystal structure of ternary InZnF_3_.

**Figure 2 materials-15-02669-f002:**
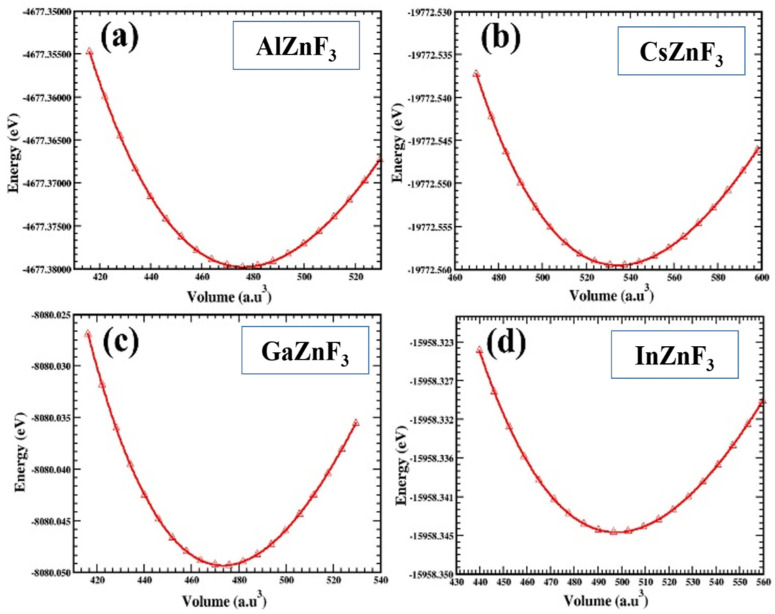
Variation in energy against volume for XZnF_3_ (X = Al, Cs, Ga, In).

**Figure 3 materials-15-02669-f003:**
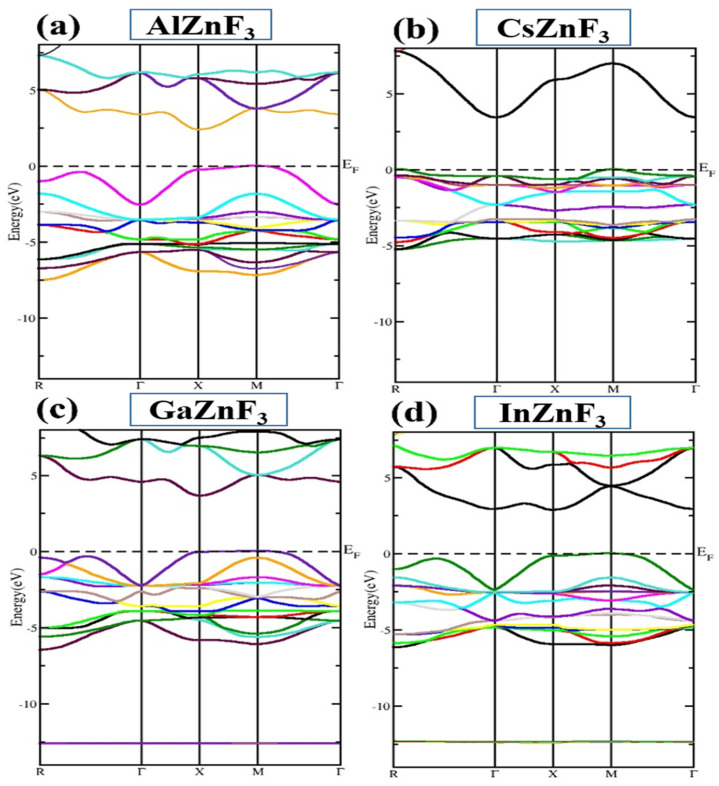
The calculated band structure of XZnF_3_ (X = Al, Cs, Ga, In). Different colors depict the various atomic states.

**Figure 4 materials-15-02669-f004:**
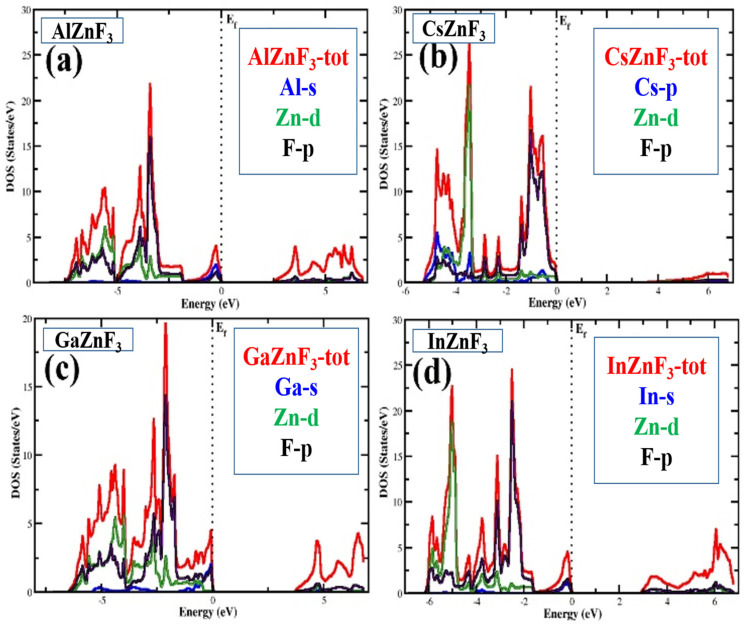
Total and partial density of states (TDOS and PDOS) of XZnF3 (X = Al, Cs, Ga, In) compounds.

**Figure 5 materials-15-02669-f005:**
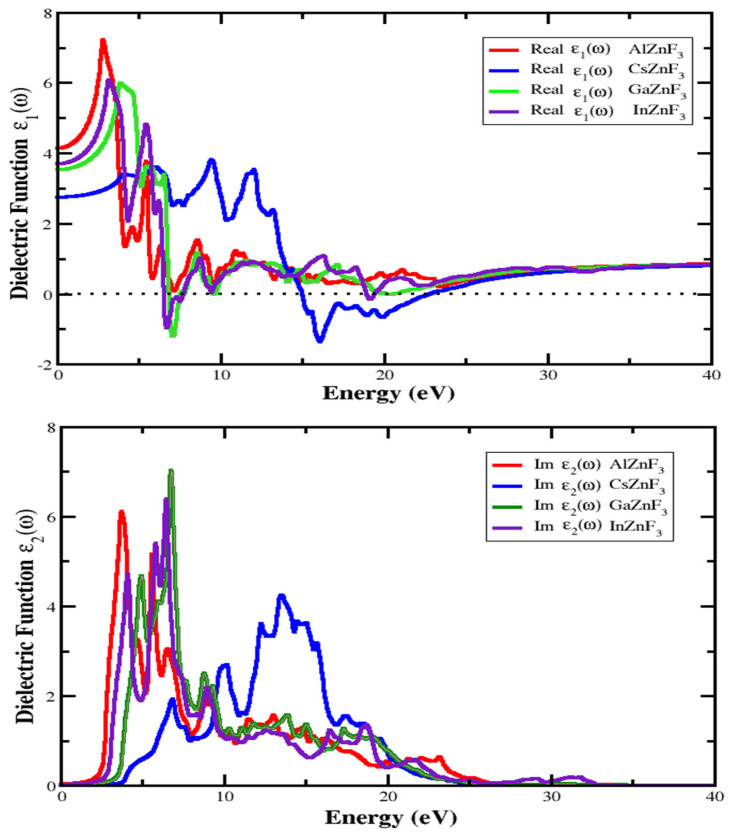
Real part $\varepsilon_{1}\left(\omega \right)$ and imaginary part $\varepsilon_{2}\left(\omega \right)$ of the dielectric function for XZnF_3_ compounds (X = Al, Cs, Ga, In).

**Figure 6 materials-15-02669-f006:**
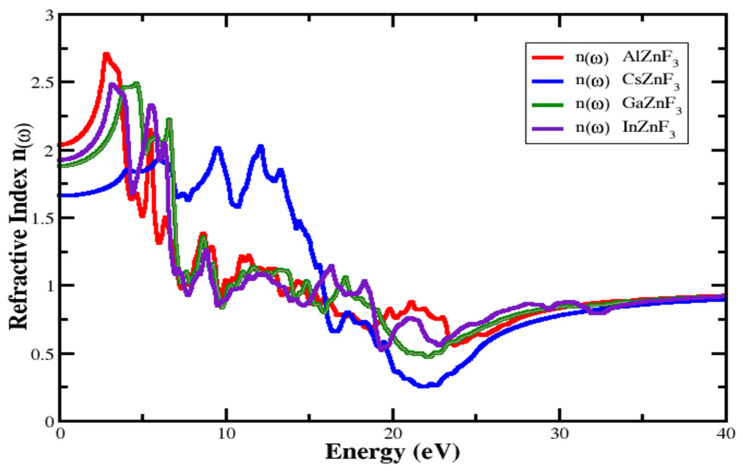
Investigated refractive index n(ω) of the dielectric function for XZnF_3_ (X = Al, Cs, Ga, In).

**Figure 7 materials-15-02669-f007:**
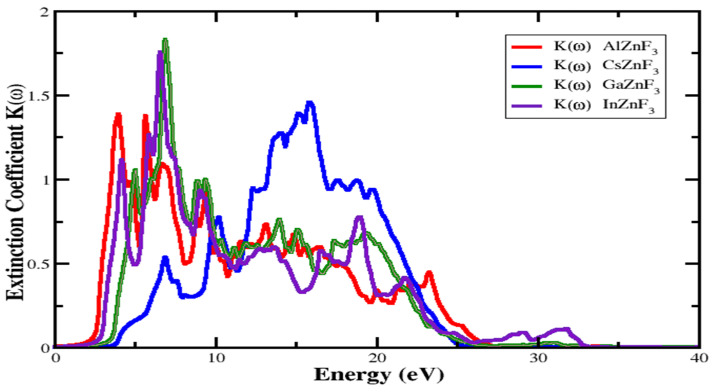
Investigated extinction coefficient K(ω) of the dielectric function for XZnF_3_ (X = Al, Cs, Ga, In).

**Figure 8 materials-15-02669-f008:**
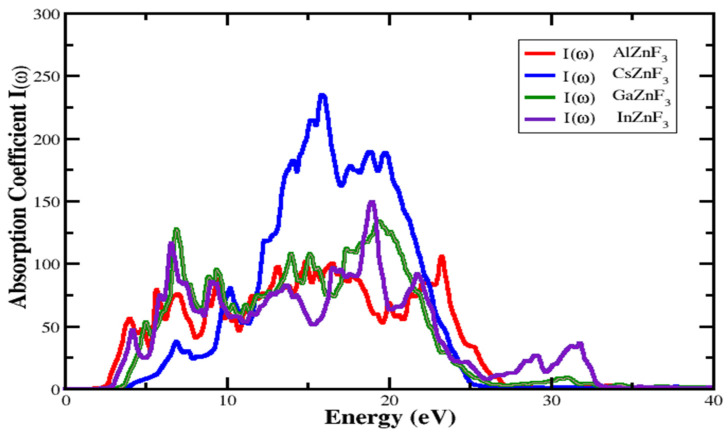
Absorption coefficient of XZnF_3_ (X = Al, Cs, Ga, In).

**Figure 9 materials-15-02669-f009:**
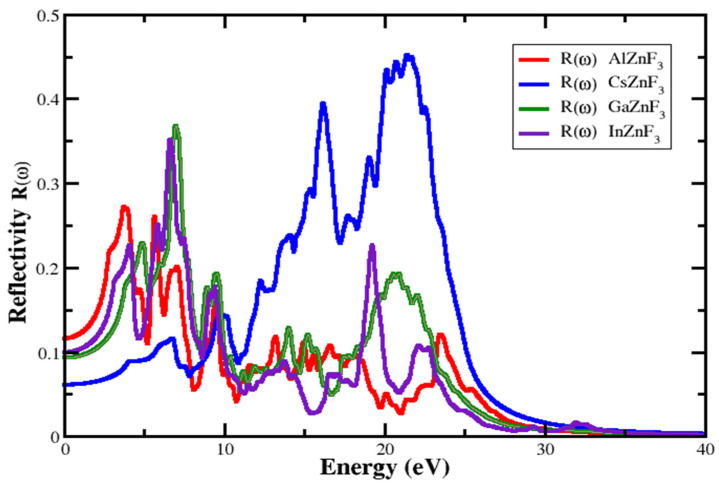
Reflectivity of XZnF_3_ (X = Al, Cs, Ga, In).

**Figure 10 materials-15-02669-f010:**
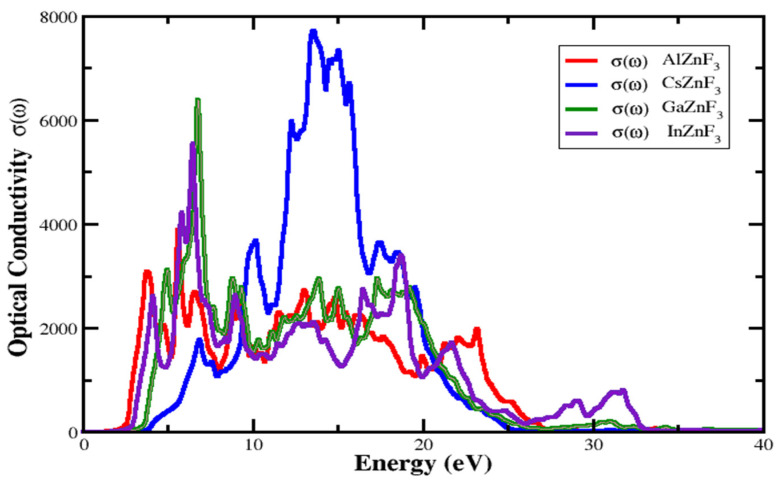
Optical conductivity of XZnF_3_ (X = Al, Cs, Ga, In).

**Table 1 materials-15-02669-t001:** Using GGA potential, calculated values of structural parameters include ground state lattice constant (a_0_), ground state volume (V_0_), ground-state energy (E_0_), bulk modulus B(GPa), and the derivative of bulk modulus $B^{\prime }$ of XZnF_3_ compounds (X = Al, Cs, Ga, In).

Compounds	Lattice Constant (a_0_)	Bulk Modulus (GPa)	Derivative of Bulk Modulus (GPa)	Ground State Volume (V_0_)	Ground State Energy (E_0_)
CsZnF3	4.29 Å	64.74	4.86	534.40	−19,772.55
InZnF3	4.19 Å	72.28	4.71	497.66	−15,958.34
GaZnF3	4.12 Å	74.57	4.56	473.29	−8080.04
AlZnF3	4.13 Å	74.66	4.56	476.24	−4677.37

**Table 2 materials-15-02669-t002:** Calculated values for the elastic constants of XZnF_3_ (X = Al, Cs, Ga, In). The computed cubic elastic constants C_11_, C_12_, C_44_, the bulk modulus B, anisotropy factor A, Young’s modulus E, Poisson ration υ, Shear modulus G, and the Pugh ratio B/G.

Compounds	C_11_ (GPa)	C_12_ (GPa)	C_44_ (GPa)	B (GPa)	A	G (GPa)	E (GPa)	v	B/G
CsZnF_3_	87.43	53.96	35.36	65.13	2.11	26.19	69.28	0.45	2.48
InZnF_3_	90.94	61.18	17.95	71.00	1.20	16.65	46.34	0.56	4.26
GaZnF_3_	100.18	60.45	21.45	73.70	1.08	20.80	53.04	0.53	3.54
AlZnF_3_	95.22	66.82	19.01	76.45	1.33	16.91	47.26	0.57	4.51

## Data Availability

Not applicable.

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
