# Peer review of "Insight into the Exemplary Physical Properties of Zn-Based Fluoroperovskite Compounds XZnF3 (X = Al, Cs, Ga, In) Employing Accurate GGA Approach: A First-Principles Study"

_materials, 2022, doi:10.3390/ma15072669_

Round 1
Reviewer 1 Report
This paper demonstrated that the structural, elastic, electronic and optical properties implying DFT for the cubic fluoroperovskites XZnF3 (X = Cs, In, Ga, Al) compounds. Mechanical stability and electronic properties were calculated through theoretical calculations. However, there seems to be a lack of application on how these calculations can be considered in the electronic and semiconductor processing industries. In addition, even a theoretical study requires a detailed explanation of the experimental(theoretical) method and analysis.
Comment to reinforce the manuscript.
- The table style varies.
- In Figure 1 and 3, there are no figure legend.
- In figure 2, 3, and 4, the text in the figures is too small.
- It requires more detailed explanation about the method and analysis for each experiment(simulation).
- Cs is a group 1 element and has different characteristics from group 13 elements Al, Ga, and In. Therefore, as shown in Figure 5-8, it seems natural that the electrical characteristics appear differently. At the conclusion, the author mentioned : Electronic properties of these compounds reveals that all are semiconductors having an indirect band gaps in CsZnF3 from (?? − ? ), while InZnF3, GaZnF3 and AlZnF3 have indirect band gap (?? − ?? ). We have found bonding character for these compounds that is dominantly ionic, and partly covalent. However, for chemist, It seems natural that Cs is a group 1 element with strong ionic bonding.
Author Response
Response to Reviewer 1:
[1] We thank the reviewer for such a nice comment. “The table style varies.”
Upon the recommendation of the reviewer we have made the style of the tables the same.
Table 3.1. Using GGA potential, calculated values of structural parameters just like ground state lattice constant (a0), ground state volume (V0), and ground state energy (E0), bulk modulus B(Gpa), derivative of bulk modulus of XZnF3 compounds (X = Al, Cs, Ga, In).
Compounds |
Lattice Constant (a0) |
Bulk Modulus (GPa) |
Derivative of Bulk Modulus (GPa) |
Ground State Volume (V0) |
Ground State Energy (E0) |
CsZnF3 |
4.294 |
64.744 |
4.867 |
534.406 |
-19772.559 |
InZnF3 |
4.194 |
72.284 |
4.718 |
497.667 |
-15958.345 |
GaZnF3 |
4.124 |
74.574 |
4.569 |
473.293 |
-8080.049 |
AlZnF3 |
4.133 |
74.665 |
4.563 |
476.241 |
-4677.379 |
Table 3.2. Calculated values for the elastic constants of XZnF3 (X = Al, Cs, Ga, In). The computed cubic elastic constants C11, C12 , C44 , the bulk modulus B, anisotropy factor A, Young’s modulus E, Poisson ration Ê‹, Shear modulus G, and the Pugh ratio B/G.
Compounds |
C11 (GPa) |
C12 (GPa) |
C44 (GPa) |
B (GPa) |
A |
G (GPa) |
E (GPa) |
v |
B/G |
CsZnF3 |
87.437 |
53.966 |
35.366 |
65.136 |
2.113 |
26.191 |
69.288 |
0.457 |
2.487 |
InZnF3 |
90.948 |
61.185 |
17.958 |
71.002 |
1.207 |
16.657 |
46.347 |
0.566 |
4.263 |
GaZnF3 |
100.182 |
60.454 |
21.453 |
73.702 |
1.080 |
20.803 |
53.043 |
0.533 |
3.542 |
AlZnF3 |
95.220 |
66.823 |
19.015 |
76.453 |
1.339 |
16.916 |
47.262 |
0.576 |
4.519 |
[2] We thank the reviewer for such a nice comment. “In Figure 1 and 3, there are no figure legend.”
Following the kind suggestion of the reviewer, we have added the legends in Figure 1 and 3.
Figure 1. Structure of XZnF3 (X = Al, Cs, Ga, In).
Figure 3. The calculated band structure of XZnF3 (X = Al, Cs, Ga, In).
[3] We thank the reviewer for such a nice comment. “In figure 2, 3, and 4, the text in the figures is too small.”
Upon the suggestion of the reviewer we have changed the text size in the figure 2, 3 and 4.
Figure 2. Variation in energy against volume for XZnF3 (X= Al, Cs, Ga, In).
Figure 3. The calculated band structure of XZnF3 (X = Al, Cs, Ga, In).
Figure 4. Total and partial density of states (TDOS and PDOS) of XZnF3 (X = Al, Cs, Ga, In) compounds.
[4] We thank the reviewer for such a nice comment. “It requires more detailed explanation about the method and analysis for each experiment (simulation).”
Following the kind suggestion of the reviewer, the more detail explanation about the method and analysis for each simulation is added which as highlighted as yellow in the manuscript.
For structural optimization, integral through the Brillouin zone is done taking 2000 k-points from the mesh in the full Brillouin zone. The structural properties are simulated using the Birch-Murnaghan equation of state by fitting the energy versus volume curve of the crystal unit cell. The electronic properties are investigated using the GGA approximation within the high symmetries points of the first Brillouin zone. The DOS is reported within the energy ranges from -8 eV to 8 eV. For the computation of elastic properties the IRelast package developed by J. Murtaza is employed for elastic constants and other parameters. The optical properties for all the interested compounds are investigated using the dielectric function within the energy range from 0 eV up to 40 eV. Based on the aforementioned computational methods, we found the very accurate and precise results as are describe below.
[5] We thank the reviewer for such a nice comment. “Cs is a group 1 element and has different characteristics from group 13 elements Al, Ga, and In. Therefore, as shown in Figure 5-8, it seems natural that the electrical characteristics appear differently. At the conclusion, the author mentioned: Electronic properties of these compounds reveals that all are semiconductors having an indirect band gaps in CsZnF3 from (?? − ?), while InZnF3, GaZnF3 and AlZnF3 have indirect band gap (?? − ??). We have found bonding character for these compounds that is dominantly ionic, and partly covalent. However, for chemist, it seems natural that Cs is a group 1 element with strong ionic bonding.”
We are fully agree with the author comment. The basic focus of presenting the above sentence in the manuscript is that it bears the idea of interacting alkali metals and transition metals with ZnF3.

Reviewer 2 Report
The manuscript present properties of a new fluoroperovskites XZnF3 material, before the publication I recomend a minor revisions.
Improve the discussion and comparison with published data from other similar materials.
Author Response
Response to Reviewer 2:
[6] We thank the reviewer for appreciating us and for such a nice comment. “Improve the discussion and comparison with published data from other similar materials.”
Following the kind suggestion of the reviewer we have significantly improved the discussion part and a comparative study is done from published data for others similar materials.
The similar results for the structural properties is reported by Husain et. al. while in investigating the different physical properties of the NaQF3 (Q = Ag, Pb, Rh, and Ru) compounds. The structural stability, bulk modulus, lattice constants, and other ground state parameters were reported by them. Based on the comparison of the interested compounds with other similar type of fluoroperovskites, we can confidently declare that our selected materials have a stable cubic crystal structure. All the data presented in this manuscript are compared wherever is needed for the precision and accuracy purpose.
Reviewer 3 Report
Despite the introduction, which describes the relevance of studying Zn-based fluoroperovskites compounds well, the article itself and the interpretation of the results are unsatisfactory.
- The article has serious remarks on the language. The style should be changed almost everywhere. Here is an example of the first paragraph (corrections are highlighted in yellow, places that are stylistically problematic are highlighted in red). Unfortunately, the entire text has serious stylistic errors and should be completely revised.
An example: Among various compounds, one of the families of compounds known as perovskites has drawn the attention of the material scientists because of having a significant applications (style!) in the field of science and technology. These kinds of materials are studied by the researchers using computational and experimental approaches. The compound (CaTiO3) which is a mineral perovskite was firstly investigated in Ref [1]. After this an incredible attention was given to this family of compounds. Because of having the effective uses in the science and technology extraordinary concentration was set to optical as well as electronic properties of perovskites in the industry of lenses and semiconductors [2].
2) Table headings and figure captions are incomplete. For example, the caption of Table 3.2. sounds as „Calculated values for the elastic constants of XZnF3 (X = Al, Cs, Ga, In)”. What physical quantities are considered in the table? This should be commented on.
Captions for pictures and tables must be independent and contain complete information that is indicated in the figure or in the table.
3) Many spaces are missed in the text.
E.g. «At about23.32 eV for AlZnF3, 19.460 eV for CsZnF3, 19.2 eV for GaZnF3and around18.97eV for InZnF3the maximum I(ω) is calculated to be about 104.832, 234.034, 133.71, and 149.47 respectively»
«The higher?(?)is found out to be3869.75 Ω-1 cm-1 at 5.674 eV for AlZnF3, 7725.49Ω-1 cm-1 at 13.578 eV for CsZnF3, 6577.03Ω-1 cm-1 at6.899 eV is for GaZnF3 and 5834.73Ω-1 cm-1 at 6.654 eV is found for AlZnF3».
4) Figure 1 depicting a cubic perovskite is flattened.
5) The colors in figure 3 are not explained
6) Style in the bibliography list is patchy
7) The interpretation of the obtained values is not clear from the article. At the moment it looks like a set of numbers generated by the program. Without proper interpretation, they are of no scientific interest.
In my opinion, an article of this quality cannot be published in a journal with an impact factor higher than 3. I cannot offer the editors a complete revision, since the article should not only be completely rewritten due to shortcomings in the English language. Unfortunately, a complete reworking of the interpretation of the results is also needed. If the interpretation of the results is improved for resubmitted version, then it should be reassessed whether the work has scientific merit.
Author Response
Response to Reviewer 3:
[1] We thank the reviewer for such a nice comment. “The article has serious remarks on the language. The style should be changed almost everywhere. Here is an example of the first paragraph (corrections are highlighted in yellow, places that are stylistically problematic are highlighted in red). Unfortunately, the entire text has serious stylistic errors and should be completely revised”.
Following the kind suggestion of the reviewer we have greatly improved the language of the manuscript. We have gone through the whole manuscript very precisely and fitted the stylistically problems and typos mistakes. Hopefully the manuscript now bears well and the readability is greatly improved.
[2] We thank the reviewer for such a nice comment. “Table headings and figure captions are incomplete. For example, the caption of Table 3.2. Sounds as „Calculated values for the elastic constants of XZnF3 (X = Al, Cs, Ga, In)”. What physical quantities are considered in the table? This should be commented on.” Captions for pictures and tables must be independent and contain complete information that is indicated in the figure or in the table.
Following the kind suggestion of the reviewer we have precisely checked and completed the Table headings and figure captions wherever is needed. The captions for plots and table are now independent and contain complete information that is indicated in figure or table.
Table 3.2. Calculated values for the elastic constants of XZnF3 (X = Al, Cs, Ga, In). The computed cubic elastic constants C11, C12 , C44 , the bulk modulus B, anisotropy factor A, Young’s modulus E, Poisson ration Ê‹, Shear modulus G, and the Pugh ratio B/G.
Compounds |
C11 (GPa) |
C12 (GPa) |
C44 (GPa) |
B (GPa) |
A |
G (GPa) |
E (GPa) |
v |
B/G |
CsZnF3 |
87.437 |
53.966 |
35.366 |
65.136 |
2.113 |
26.191 |
69.288 |
0.457 |
2.487 |
InZnF3 |
90.948 |
61.185 |
17.958 |
71.002 |
1.207 |
16.657 |
46.347 |
0.566 |
4.263 |
GaZnF3 |
100.182 |
60.454 |
21.453 |
73.702 |
1.080 |
20.803 |
53.043 |
0.533 |
3.542 |
AlZnF3 |
95.220 |
66.823 |
19.015 |
76.453 |
1.339 |
16.916 |
47.262 |
0.576 |
4.519 |
[3] We thank the reviewer for such a nice comment. “Many spaces are missed in the text”.
Following the kind suggestion of the reviewer we have precisely gone through the whole manuscript and fixed the spaces which are missed in the text wherever is needed.
[4] We thank the reviewer for such a nice comment. “Figure 1 depicting a cubic perovskite is flattened”.
Following the kind suggestion of the reviewer we have re-insert the Figure 1 depicting a cubic perovskite.
Figure 1. Structure of XZnF3 (X = Al, Cs, Ga, In).
[5] We thank the reviewer for such a nice comment. “The colors in figure 3 are not explained”.
Following the kind suggestion of the reviewer and we respect his comment regarding the explanation of different colors in figure 3.
Figure 3. The calculated band structure of XZnF3 (X = Al, Cs, Ga, In).
The existence of different colors in the above figure 3 just shows the contributions of different atomic states of different constituent atoms of the compound to the valence and conduction band. Depiction of different colors corresponds to different atomic states contributions.
[6] We thank the reviewer for such a nice comment. “Style in the bibliography list is patchy”.
This is very nice comment, we fully respect the reviewer comment. The bibliography list in in IEEE style, which is in accord to the journal requirements.
[7] We thank the reviewer for such a nice comment. “The interpretation of the obtained values is not clear from the article. At the moment it looks like a set of numbers generated by the program. Without proper interpretation, they are of no scientific interest”.
This is very nice comment from the respected reviewer. We have reviewed again the interpretation of the obtained values in the manuscript wherever is needed. All the values computed from the simulation are described and interpreted now in a very scientific way. Hopefully the values we have are of scientific interest.

Round 2
Reviewer 1 Report
It is well revised
Author Response
Thanks for appreciations.
Reviewer 3 Report
The paper still requires complete revision before the re-submission.
1) there are still grammatical and stylistic errors in the text (their number has decreased compared to the first version, but the manuscript needs proofreading). See some examples in the attached pdf.
2) Picture 1 requires a different caption. Either XZnF3 is depicted (then the picture needs to be changed), or InZnF3 is depicted (and then the caption needs to be changed). The picture is still deformed and the compound does not look cubic.
3) physical units are not correctly written everywhere (Gpa -> GPa)
4) colors in Figure 3 are still not commented
> The existence of different colors in the above figure 3 just shows the contributions of different atomic states of different constituent atoms of the compound to the valence and conduction band. Depiction of different colors corresponds to different atomic states contributions.
This clarification should be inserted into the text / caption of the figure
5) list of literature contains additional empty lines (see distance between Ref. 12 and 13, Ref. 24 and Ref. 25 etc). In some Refs. the full list of authors is used on the other hands in other Refs. only part of authors is mentioned. (e.g. “K. V Kravchyk, T. Zünd, M. Wörle, M. V Kovalenko, and M. I. Bodnarchuk”, Ref. 12, vs “G. Vaitheeswaran et al.” Ref. 9).
6) Table 3.2 caption is written with another font. The captions of both Tables differ (please, write physical units for all quantities presented in the tables). E.g. Units of Lattice Constant aren’t given in the text.
7) From Reviewing 1st round: The interpretation of the obtained values is not clear from the article. At the moment it looks like a set of numbers generated by the program. Without proper interpretation, they are of no scientific interest”.
> This is very nice comment from the respected reviewer. We have reviewed again the interpretation of the obtained values in the manuscript wherever is needed. All the values computed from the simulation are described and interpreted now in a very scientific way. Hopefully the values we have are of scientific interest.
The main disadvantage relates to the lack of comparison with experimental work or the absence of significant theoretical conclusions. Without this, the work cannot be published in a journal with a high impact factor.
8) Calculation errors are also not evaluated. Three decimal places accuracy is usually unphysical in computational science.

Author Response
We thank the reviewer for such a nice comment.
- Following the reviewer suggestion the following changes are made.
In perovskite materials all types of compounds such as conductor, semiconductor, and insulator exist.
Additionally, the ability of fluoroperovskites materials in numerous scientific applications as a fuel for cells, memory appliances, photovoltaic etc. are studied and has been demonstrated to be outstanding compounds for microelectronics as well as in telecommunications.
To obtain the total energy convergence, the basic functions in the IR are expanded up to Rmt × Kmax = 6.0 inside the atomic spheres for the wave function.
It could be seen from the table that the C11 value for GaZnF3 is greater which indicates that it is most elastic than that of other materials.
The optical properties of XZnF3 compounds (X = Al, Cs, Ga, In) are evaluated by using the GGA+U method within the Density Functional theory scheme.
It may be seen from fig 5 that the compounds are more effective in the range of energy of 2.5 eV to 9 eV in the case of while in the case of compounds are more active in between 4 eV to 14 eV
The electronic properties of these compounds reveal that the compounds of interest are semiconductors having an indirect bandgap in CsZnF3 from (?? − ?), while InZnF3, GaZnF3, and AlZnF3 have indirect bandgap (?? − X).
We are fully confident in our more precise results and the applications of the above-reported compounds can be deemed in many electronics and semiconducting processing industries.
- We thank the reviewer for such a nice comment. Following your kind suggestion the caption for figure 1 is changed now and as is following.
Figure 1. Prototype crystal structure of ternary InZnF3.
The picture is placed is the one taken from the software and is not deformed, and is purely cubic structure.
- We thank the reviewer for such a nice comment.
The unit GPa is corrected throughout the manuscript and is written as GPa.
- We thank the reviewer for such a nice comment. Caption for Fig. 3 is changed as following
Figure 3. The calculated band structure of XZnF3 (X = Al, Cs, Ga, In). Different colors depicts the various atomic states.
- We thank the reviewer for such a nice comment. The difference among references 12, 13, 9, 24 and 25 are removed and are made similar.
[9] G. Vaitheeswaran, V. Kanchana, Ravhi S. Kumar, A. L. Cornelius, M. F. NiCol, A. Svane, N. E. Christensen, O. Eriksson "High-pressure structural study of fluoro-perovskite CsCdF3 up to 60 GPa: A combined experimental and theoretical study," Phys. Rev. B, vol. 81, no. 7, pp. 0751051-0751056, 2010.
[12] K. V Kravchyk, T. Zünd, M. Wörle, M. V Kovalenko, and M. I. Bodnarchuk, “NaFeF3 nanoplates as low-cost sodium and lithium cathode materials for stationary energy storage,” Chem. Mater., vol. 30, no. 6, pp. 1825–1829, 2018.
[13] R. Arar, T.Ouahrani, D.Varshney, R.Khenata, G. Murtaza, D. Rached, A.Bouhemadou, Y.Al-Douri, S. BinOmran, A. H. Reshak., "Structural, mechanical and electronic properties of sodium based fluoroperovskites NaXF3 (X= Mg, Zn) from first-principle calculations," Mater. Sci. Semicond. Process., vol. 33, pp. 127-135, 2015.
Reference 24 contains only two authors and reference 25 contains only one author.
- We thank the reviewer for such a nice comment. The captions of both the tables are now same. Units of lattice constants are inserted as Å in the main text.
- We thank the reviewer for such a nice comment. “The interpretation of the obtained values is not clear from the article. At the moment it looks like a set of numbers generated by the program. Without proper interpretation, they are of no scientific interest”.
This is very nice comment from the respected reviewer. We have reviewed again the interpretation of the obtained values in the manuscript wherever is needed. All the values computed from the simulation are described and interpreted now in a very scientific way. Hopefully the values we have are of scientific interest. For instance, ground state volume, ground state energy, bulk modulus and derivative of bulk modulus helps in explain the stability of the studied compounds. Similarly, the obtained lattice constants demonstrates the mechanical stability. Each obtained value bears scientific interpretation and is discussed in the main article.
- We thank the reviewer for such a nice comment. Calculations error are also evaluated and is kept only up to two decimal places.

This manuscript is a resubmission of an earlier submission. The following is a list of the peer review reports and author responses from that submission.